# A Triphenylphosphonium-Functionalized Delivery System for an ATM Kinase Inhibitor That Ameliorates Doxorubicin Resistance in Breast Carcinoma Mammospheres

**DOI:** 10.3390/cancers15051474

**Published:** 2023-02-25

**Authors:** Venturina Stagni, Archontia Kaminari, Claudia Contadini, Daniela Barilà, Rosario Luigi Sessa, Zili Sideratou, Spiros A. Vlahopoulos, Dimitris Tsiourvas

**Affiliations:** 1Institute of Molecular Biology and Pathology, National Research Council (CNR), 00183 Rome, Italy; 2Laboratory of Cell Signaling, IRCCS, Fondazione Santa Lucia, 00179 Rome, Italy; 3Institute of Nanoscience and Nanotechnology, NCSR ‘‘Demokritos”, 15310 Aghia Paraskevi, Greece; 4Department of Biology, University of Tor Vergata, 00133 Rome, Italy; 5Horemeio Research Laboratory, First Department of Paediatrics, N.K.U.A, 11527 Athens, Greece

**Keywords:** ataxia-telangiectasia mutated kinase, cancer stem cells, triphenylphosphonium cation

## Abstract

**Simple Summary:**

Doxorubicin (DOX) is widely used in the treatment of breast cancer. However, resistance limits its effectiveness. In particular, breast cancer stem cells (BCSCs) are associated with DOX resistance. We have previously demonstrated the potential of a polymeric nanocarrier based on a suitably functionalized hyperbranched polyethylenimine that preferentially targets BCSCs. ATM kinase is a key mediator of DNA damage response, so its inhibition has become an attractive therapeutic concept in cancer therapy for the sensitization of cancer cells to chemotherapeutic drugs. Herein, we tested the potential of this drug delivery system that encapsulates an ATM inhibitor to target and sensitize mammospheres—considered as a model system of BCSCs—to an anticancer drug, while having a comparably lower cytotoxic effect against bulk tumor cells.

**Abstract:**

The enzyme ataxia-telangiectasia mutated (ATM) kinase is a pluripotent signaling mediator which activates cellular responses to genotoxic and metabolic stress. It has been shown that ATM enables the growth of mammalian adenocarcinoma stem cells, and therefore the potential benefits in cancer chemotherapy of a number of ATM inhibitors, such as KU-55933 (KU), are currently being investigated. We assayed the effects of utilizing a triphenylphosphonium-functionalized nanocarrier delivery system for KU on breast cancer cells grown either as a monolayer or in three-dimensional mammospheres. We observed that the encapsulated KU was effective against chemotherapy-resistant mammospheres of breast cancer cells, while having comparably lower cytotoxicity against adherent cells grown as monolayers. We also noted that the encapsulated KU sensitized the mammospheres to the anthracycline drug doxorubicin significantly, while having only a weak effect on adherent breast cancer cells. Our results suggest that triphenylphosphonium-functionalized drug delivery systems that contain encapsulated KU, or compounds with a similar impact, are a useful addition to chemotherapeutic treatment schemes that target proliferating cancers.

## 1. Introduction

Resistance to genotoxic therapies has been associated with increased DNA damage response (DDR) signaling, and many cancer defects in certain components of the DDR are highly dependent on the remaining DDR pathways for survival [1]. The enzyme ataxia-telangiectasia mutated (ATM) kinase is a key mediator of DDR, and its inhibition has become an attractive therapeutic concept in cancer therapy for the sensitization of cancer cells to chemotherapeutic drugs. In particular, cancer stem cells (CSCs) have been gathering increasing attention over the past decade as they play a crucial role in tumor progression, metastasis, and drug resistance [2].

It has been well documented that ATM kinase inhibition sensitizes cells to the cytotoxic effects of DNA double-strand break-inducing chemotherapeutic agents, including the topoisomerase II inhibitors etoposide, doxorubicin, and amsacrine, the topoisomerase I inhibitor camptothecin, and PARP inhibitors [3,4]. Furthermore, the ATM inhibitor KU-55933 (KU) was shown to sensitize p53-deficient cholangiocarcinoma cells to genotoxic agents, including gemcitabine, 5-fluorouracil, cisplatin, and doxorubicin. This sensitization was more potent when KU was combined with ATR (ataxia-telangiectasia mutated and Rad3-related kinase) inhibitor VE-821 [5]. KU has also been reported to block the phosphorylation of protein kinase B (Akt) and inhibit MDA-MB-453 and PC-3 cell proliferation [6], as well as to attenuate the phosphorylation and activation of AMP-activated protein kinase in a rat hepatoma cell line [7]. The inhibition of Akt was confirmed and extended when it was shown that glucose uptake, glycolysis, epithelial to mesenchymal transition, motility, and the proliferation of aggressive breast and prostate cancer cell lines with high Akt activity were blocked by KU [8].

Interestingly, ATM kinase is also an essential signaling mediator that enables the growth of cancer stem cells. Thus, the potential benefits of a number of ATM inhibitors, such as KU-55933 (KU), in overcoming the resistance of CSCs to chemotherapeutic agents are currently being investigated [9,10,11,12]. It has been found that the application of this ATM inhibitor effectively decreases the radiation resistance of the tumorspheres of cancer initiating cells, which are stem-like cells [13]. In fact, ATM is known to sustain the mammospheres of cancerous cells and are considered model CSC systems that are also known to exhibit high resistance to genotoxic agents, such as doxorubicin (DOX) [14,15].

ATM function is linked to mitochondria. It is well known that it regulates mitochondrial function and mitophagy [16,17,18,19]. In this connection, studies on the action of KU on cell lines have shown that its administration reduces the mitochondrial membrane potential and perturbs the tricarboxylic acid (TCA) cycle and oxidative phosphorylation [20,21,22,23]. KU has also been shown to suppress the proliferation of Hep G2 and SMMC-7721 cells by inducing mitochondrial dysfunction and by enhancing 5′-adenosine monophosphate-activated protein kinase (AMPK) phosphorylation [21].

Nutlin-3 (an MDM2 inhibitor that leads to non-genotoxic p53 activation) and KU synergize to induce apoptosis in a number of cancer cell types, including colorectal cancer cell lines, but do not kill non-transformed cells. The mechanism of cell death activation entails the blocking of autophagy and a consequent accumulation of both mitochondria and reactive oxygen species (ROS) [22]. It has also been reported that the inhibition of ATM with KU depleted mitochondrial DNA in wild-type fibroblasts [23]. Therefore, ATM suppression would be advantageous in eradicating cancer cells, in particular CSCs, but disadvantageous for normal cells because its function is indispensable in DNA repair, in preserving mitochondrial functionality, and in the selective removal of damaged mitochondria [17]. Consequently, there is a need of a delivery system for ATM inhibitors that, ideally, specifically targets cancer stem cells.

The essential role of mitochondria in cell function and the fact that mitochondrial dysfunctions are associated with a number of pathologies, including cancer, had led to the growth of so-called mitochondrial medicine and, in parallel with this, to the development of systems capable of the accurate and efficient delivery of therapeutic agents and/or imaging agents to mitochondria. A number of mitochondriotropic moieties, i.e., moieties that can target mitochondria, have already been identified, from mitochondria-penetrating peptides to delocalized lipophilic cations, such as the well-studied triphenylphosphonium cation (TPP). In the latter case, mitochondrial internalization is caused by the delocalized positive charge of the TPP and the large negative membrane potential of the mitochondria (Δ*Ψ*_m_ = 150–180 mV) [24,25]. Mitochondrial targeting by drugs has been achieved by employing these mitochondriotropic agents, either by their direct conjugation with bioactive molecules or by their conjugation with a variety of drug-loaded delivery systems [26,27,28,29,30,31].

Previous studies by our group have indicated that TPP-functionalized hyperbranched polyethylenimine nanoparticles (PTPP) can encapsulate DOX and target mitochondria, causing severe cytotoxicity at low DOX concentrations [32]. Interestingly, this nanocarrier also showed selective cytotoxicity against mammospheres that depended on the expression of the gene encoding ATM kinase [33]. In this context, it has been reported that the enhanced tolerance of CSCs to chemotherapeutics or radiation correlates well with the changes in membrane potential, and that cells with higher membrane potential are more prone to continue dividing and form tumors compared with cells with lower membrane potentials [34]. Therefore, TPP-functionalized carriers may also have the ability to target cells with high membrane potential, such as CSCs [33,34]. Indeed, TPP-functionalized nanoparticles have shown that cell internalization is dependent on cell membrane potential, which enables them to be internalized preferably to cancerous but not to non-cancerous cells [35,36]. In this current study, we explore whether encapsulating an ATM inhibitor in a TPP-functionalized hyperbranched polyethylenimine nanocarrier can be effective against the mammospheres of drug resistant breast cancer cells—a close analogue to CSCs—without having considerable toxic effects against adherent cells. We further explore the resistance of mammospheres derived from breast cancer cell lines against DOX chemotherapy and the therapeutic benefit of ATM inhibition upon administration of the water insoluble ATM kinase inhibitor KU, encapsulated in the PTPP nanocarrier (PTPP–KU).

## 2. Materials and Methods

### 2.1. Chemicals and Reagents

RPMI-1640 medium, penicillin/streptomycin, L-glutamine, phosphate buffer saline (PBS), and trypsin/EDTA were all purchased from Biochrom (Berlin, Germany), while HyClone fetal bovine serum was obtained from Invitrogen (Carlsbad, CA, USA). Dimethyl sulfoxide (DMSO) and MTS solution were purchased from Merck KGaA (Calbiochem^®^, Darmstadt, Germany) and Promega Corp. (Madison, WI, USA), respectively, while 2-Morpholin-4-yl-6-thianthren-1-yl-pyran-4-one (KU-55933) was obtained from Sigma-Aldrich Ltd. (Poole, UK). Doxorubicin (D1515) was purchased from Sigma-Aldrich Corp. (St. Louis, MO, USA).

### 2.2. Preparation and Characterization of KU-Loaded PTPP Nanoparticles

The introduction of decyltriphenylphosphonium groups to hyperbranched polyethylenimine (PTPP) has been detailed in our previous publications [32,33]. Given that both PTPP and KU are practically water insoluble, a co-precipitation method was established for either the formation of PTPP nanoparticles or the encapsulation of KU in the PTPP nanoparticles, entailing the drop-wise addition of 100 μL DMSO solution of PTPP and KU (KU concentration 7 mΜ, PTPP concentration 7 mg/mL) in 10 mL RPMI medium under vigorous stirring. In a similar manner, empty PTPP nanoparticles were prepared by the drop-wise addition of 100 μL DMSO solution of PTPP (7 μg/mL) in 10 mL RPMI under vigorous stirring. In all cases, the obtained nanoparticles were centrifuged and redispersed in the appropriate amount of RPMI for obtaining, after bath sonication for ~20 s, typical nanoparticle dispersions of 100 μM KU and of 100 μg/mL PTPP, or empty PTPP nanoparticles of the same concentration. The PTPP concentrations in their dispersions were determined by dissolving them in ethanol and registering their absorbance at 275 nm and using the respective PTTP calibration curves in ethanol. For the determination of both the PTPP and KU concentrations in the respective nanoparticle dispersions, first order derivative spectroscopy was employed. Specifically, the spectra of both compounds separately, or of their mixtures in ethanol, were acquired and processed to obtain the first derivative spectra. The wavelengths 329 nm and 278 nm were selected for the KU and PTPP determinations, respectively, as at these wavelengths there is no interference from the other compound. Similarly, the derivative spectra of standard solutions were obtained, and the respective calibration curves for each compound at these wavelengths were also derived. The mean hydrodynamic radii of the dispersions of the PTPP and PTPP–KU nanoparticles in RPMI were determined using dynamic light scattering (DLS) (AXIOS-150/EX, Triton, Hellas, 50 mW laser source at 658 nm, Avalanche photodiode detector at an angle of 90°).

### 2.3. Cell Culture and Treatments

Human breast cancer cell lines MCF-7, MDA-MB-231, and SKBR3, obtained from the American Type Culture Collection (ATCC, Manassas, VA, USA), were grown as described previously [37]. Briefly, cells were cultured in RPMI-1640 containing 2 mM L-glutamine and 1% penicillin/streptomycin and supplemented with 10% HyClone fetal bovine serum at 37 °C in a humidified CO_2_ incubator (5%). The treatment concentrations in all the experiments always refer to the KU concentration in μΜ, while the concentration of PTPP always follows the same ratio with respect to KU (i.e., KU concentration 1 mM, PTPP concentration 1 mg/mL).

### 2.4. Mammosphere Cultures

Single cell suspensions of breast cell lines MCF-7 and MDA-MB-231 were grown in ultralow attachment 6-well plates (Corning) at a density of 4000 cell/mL in mammosphere medium (Dulbecco’s modified Eagle’s medium/F-12, containing 5 μg/mL insulin (Sigma), B27 (Invitrogen), 20 ng/mL epidermal growth factor (GIBCO), 10 ng/mL basic fibroblast growth factor (GIBCO), and 0.4% bovine serum albumin (Sigma)), as described in [33]. After 10 days, the diameters of the mammospheres were measured in phase contrast pictures (ZOE) using the ImageJ software. The mammospheres (diameter > 50 μm) were counted and the efficiency of mammosphere formation was evaluated (%SFE = number of mammospheres/number of plated cells × 100). Mammosphere pellets were collected by gentle centrifugation (900 rpm, 5 min) to further analyze for protein extraction and DOX uptake.

### 2.5. MTS Assay

The cell viability of the MCF-7 and MDA-MB-231 and their derived mammospheres under normal and treatment conditions was measured using the MTS assay. In brief, cells were transferred into a single-cell suspension and plated into ultralow 96-well plates with a density of 400 cells/100 μL per well for the mammospheres, and, in adhesion conditions, in a TC-treated well at a density 1000 cells/100 μL medium. The cells were cultured in mammosphere medium [36] and treated with different doses of KU, PTPP, or PTPP–KU at 37 °C for 30 min before DOX (1 μM) was added. After 24 h, the wells were washed with RPMI, and MTS solution was added to each well and incubated at 37 °C for 3 h. Finally, the optical density (OD) was measured at a wavelength of 492 nm and the survival rates were calculated.

### 2.6. Quantification of DOX Uptake

The adherent cells and mammospheres in the 96 well plates were treated with KU (5, 10 μM), PTPP (5, 10 μg/mL), and PTPP–KU (PTPP concentrations: 5, 10 μg/mL, KU concentrations: 5, 10 μM, respectively) for 30 min, and then DOX (1 μM) was added. In addition to the control, we also had a number of wells with cells treated only with 1 μM DOX. After 3 h, the wells were washed with RPMI (without phenol red) and the DOX concentration was measured with an Infinite M200 plate reader (Tecan, Switzerland, λ_ex_ = 510 nm, λ_em_ = 580 nm). Throughout the experiment, a medium without phenol red (colorless) was used to avoid any interference with the final DOX concentration measurement.

### 2.7. Western Blotting (WB)

The cells were lysed in RIPA buffer (50 mM Tris-HCl pH 8.0, 150 mM NaCl, 1% NP40, 1 mM EGTA, 1 mM EDTA, 0.25% sodium deoxycholate) supplemented with protease and phosphatase inhibitors (Roche Diagnostic, Mannheim, Germany). Proteins were resolved by 10% SDS PAGE (about 30 μg of extract per lane was loaded) and transferred onto a nitrocellulose membrane (Protran BA83, GE Healthcare, Chicago, IL, USA) using a semi-dry system (Bio-Rad Laboratories S.r.l., Segrate, Italy). Blocking and antibody incubations were performed at room temperature in TBS containing 0.1% Tween 20 and 5% low fat milk for 1 h. The following antibodies were used: rabbit anti-PARP (Cell Signaling Technology, Danvers, MA, USA), mouse anti-ATM (Cell Signaling Technology, Danvers, MA, USA), anti-rabbit-pS15 p53 (Cell Signaling Technology, Danvers, MA, USA), and rabbit anti-Vinculin (Cell Signaling Technology, Danver, MA, USA). HRP-conjugated secondary antibodies (Bio-Rad Laboratories S.r.l., Segrate, MI, Italy) were revealed using the Clarity Western ECL Substrate (Bio-Rad Laboratories S.r.l., Segrate, Italy).

### 2.8. Statistical Analysis

Data are presented as mean ± standard deviation. All experiments were performed independently at least three times. Statistical significance for the various treatments was assessed using a Student’s *t*-test in GraphPad Prism (GraphPad Inc., La Jola, CA, USA). Significance was defined as * *p* < 0.05, ** *p* < 0.01, *** *p* < 0.001, **** *p* < 0.0001, using a Student’s *t*-test.

## 3. Results

The functionalization of hyperbranched polyethylenimine with decyltriphenylphosphonium groups endows amphiphilicity to hydrophilic macromolecules, which is the basis of their self-assembly into nanoparticles in aqueous media. Due to its lipophilic character, the water-insoluble KU resides in PTTP nanoparticles whose chemical structure encompasses both hydrophilic (i.e., ethylene imine) and hydrophobic (i.e., decyltriphenylphosphonium) entities, the properties of which endow the solubilizing (or encapsulating) properties of nanoparticles. Thus, PTTP nanoparticles encapsulating KU, which is known to be insoluble in aqueous buffers, were formed in RPMI. They had a mean hydrodynamic radius of about 60 nm, as was revealed by DLS experiments (Appendix A). For comparison, empty PTPP nanoparticles formed under the same conditions were slightly smaller, i.e., 55 nm (Appendix A). The KU loading was found to be 28.3% *w*/*w* corresponding to a KU concentration 1 mM when the PTPP concentration was 1 mg/mL.

To investigate the ability of these PTTP nanoparticles encapsulating KU to sensitize breast cancer cells to doxorubicin (DOX), we took advantage of the human breast adenocarcinoma cell line MCF-7 (luminal estrogen receptor-positive HER2-low), cultured either as adherent cells (“Adh”) or as tumor spheres (mammospheres, “MS”), which are a more suitable cellular model compared with adherent cells in that they more closely approximate the resistance to anticancer therapy of breast cancer cells [38]. Herein, we co-treated adherent MCF-7 cells and mammospheres with DOX at 1 µM and increasing concentrations of PTTP, PTTP–KU, and free KU (Figure 1A,B). Cell viability was measured as the percentage of the control treated with PTTP, PTTP–KU, or free KU, plus DOX, normalized to the same treatment without DOX (Figure 1A,B). As expected, the mammospheres were more resistant to DOX treatment compared with the adherent cells, which showed a reduction in cell viability of 30–40% compared with the non-treated (NT) cells (Figure 1A,B). We have previously reported that PTTP is effective in inhibiting the growth of cancer stem-like structures such as mammospheres [33]. It is of note that we confirmed these results, and that we also demonstrated that PTTP alone as well as PTTP–KU were slightly toxic to the adherent cells at all concentrations tested, and show considerable toxicity (ca 70%) to mammospheres (MS) only at a high concentration (10 μg/mL) (Appendix A). These results are consistent with those of previous studies which have indicated that the treatment of TPP-functionalized moieties increases cytotoxicity in cells grown in mammosphere conditions compared with cells grown in adherence conditions [33,39]. We attribute the observed toxicity to the fact that PTPP is preferentially internalized in the mitochondria of mammospheres, leading to an increase in mitochondrial stress [33]. Herein, we found that PTTP treatment can also effectively induce sensitization to DOX treatment in mammospheres, with a dose-dependent reduction in cell viability in DOX treated cells of about 40% at lower doses (1 μM) and 60% at higher doses (10 μM) (Figure 1B), compared with a reduction of only 10% at higher doses (10 μM) in adherent cells treated with DOX (Figure 1A). Interestingly, the results demonstrated that PTTP–KU is more effective in sensitizing resistant mammosphere cells to DOX, compared with adherent cells or mammospheres treated with PTTP alone (Figure 1B). In mammospheres derived from MCF-7 cells, PTTP–KU co-treatment with DOX led to a significant reduction of 50% of cell viability at 1 μM and to a reduction of 80% at 10 μM. This reduction in cell viability was evident from our observations of the morphology of the mammospheres which showed a significant reduction in mammosphere size after five days of culture following treatment at low concentrations (PTPP–KU 1µM) (Figure 2) and at higher concentrations (PTPP–KU 5µM) (Appendix A). It is of note that free KU is not able to sensitize cells to DOX treatment (Figure 1A,B), which suggests that encapsulated ATM inhibitors play a specific and functional role in sensitizing resistant breast cancer cells to anticancer therapies.

It is well known that DOX uptake in cells is correlated with the ability of drugs to induce cell toxicity inside the cell [39]. Moreover, several papers have demonstrated that mammosphere resistance to anti-cancer drugs is also correlated with an impaired uptake of drugs in the cells [40,41]. We consistently asked whether the PTPP–KU-dependent increased sensitivity to DOX treatment in the mammospheres could correlate with an increased DOX uptake in the cells. In Figure 3A,B, we quantified the DOX uptake in the cells treated with PTTP, PTTP–KU, and free KU at 5 µM and 10 µM, co-treated with 1 µM DOX (the respective raw data are shown in Appendix A). After washing the cells, we measured the DOX-dependent fluorescence emission (Figure 3A,B).

Compared with the free DOX treatment, the treatment with PTTP and PTTP–KU at 10 µM led to a 30–50% increase in DOX uptake in the adherent condition (Figure 3A). Interestingly, co-treatment with PTTP–KU at 5 µM and 10 µM led to an increase DOX uptake of about 20–50% in the mammospheres (Figure 3B) in a dose-dependent manner, compared with mammospheres treated with free DOX. Conversely, treatment with PTTP or free KU did not increase DOX uptake. These results suggest that the encapsulation of the ATM inhibitor in this specific drug delivery system leads to an increased uptake of DOX that is correlated with an increase in the sensitivity of mammospheres to DOX treatment.

To verify whether encapsulated KU efficiently inhibits ATM kinase activity, we decided to look at a well-known marker of ATM kinase activation, the autophosphorylation on serine 1981, and also at the phosphorylation of p53 at Ser 15 (phospho-S15), which is activated by DNA damage induced by DOX treatment. We performed a Western blot analysis using phospho-S1981 ATM antibodies and phosho-S15 p53-specific antibodies to investigate ATM kinase signaling activation upon DOX stimulation with or without PTTP, PTTP–KU, or free KU treatment (Figure 4 and Appendix A). As expected, and as is shown in Figure 4, the ATM kinase phosphorylation on S1981 induced by the DOX treatment was inhibited in the mammospheres derived from the MCF-7 co-treated with free KU inhibitor (Figure 4). As expected, the PTTP–KU treatment, as well as free KU treatment, was able to reduce the ATM kinase phosphorylation on S1981 induced by the DOX treatment more efficiently than PTPP alone. It is of note that PTTP also reduced ATM phosphorylation on S1981, indicating a role of this nanoparticle in modulating ATM kinase activity (Figure 4) (see Discussion section). Moreover, DOX stimulation induced phosphorylation on S15 of p53, a well-known ATM substrate. Unexpectedly, p53 phosphorylation on S15 induced by DOX stimulation was slightly reduced by treatment with all the compounds (PTPP, PTTP–KU, and free KU), and total p53 was stabilized after DOX induction in all samples, suggesting that the effect on cell viability of different treatments is independent of p53 status (see Discussion section).

We consequently looked at the PARP protein under different conditions as a marker of the cell death process (Figure 4). We could not detect a reduction in PARP protein levels after DOX treatment because, as expected, mammospheres are resistant to DOX treatment (Figure 4). Interestingly, we observed a reduction in PARP levels in samples co-treated with DOX and PTTP–KU, suggesting that only PTTP–KU is able to sensitize mammospheres to DOX treatment, according to the results obtained in Figure 1.

Since p53 has a central role in regulating the DOX sensitivity of breast cancer resistant cells [42,43], and we found that phosphorylation on S15 of p53 is also inhibited by empty PTTP nanoparticles, we wanted to clarify the role of p53 in the sensitization of mammospheres to DOX treatment. To evaluate the necessity for intact p53 function, we also utilized cell lines with mutant p53. We therefore performed viability experiments in mammospheres derived from p53 WT cells (MCF-7) or mutant p53 cells (SKBR3 and MDA-MB-231 cells). We co-treated the cells with DOX (1 μM) and with increasing doses of PTTP, PTTP–KU, or free KU (Figure 5). Interestingly, PTPP–KU sensitized the mammospheres derived from all the above cell lines to DOX in a dose-dependent manner, suggesting that the role of PTPP–KU in sensitizing resistant mammospheres to DOX is independent of p53 status.

## 4. Discussion

Doxorubicin is a tetracycline antibiotic commonly used in the treatment of breast cancer, and it induces DNA damage by the inhibition of topoisomerase II and free radical generation as an anticancer mechanism [44,45]. Doxorubicin has severe side effects, including acute toxicity to normal tissue and cardiotoxicity, and its therapeutic effects can be minimized by the inherent multidrug resistance (MDR) of many tumor cells, in particular of breast cancer stem cells [45,46]. The MDR of breast cancer stem cells is a major challenge to successful chemotherapy, and mitochondria-targeting therapy represents a promising strategy that may enable us to overcome MDR [47]. There are various mechanisms associated with MDR, often involving acquired and intrinsic resistance. Unlike acquired MDR, which mechanism was originated from the overexpression of P-glycoprotein (P-gp), an ATP-dependent efflux pump, intrinsic MDR is often attributed to genetic or epigenetic changes which perturb the apoptosis signaling pathway [48,49,50]. Generally, the intrinsic pathway of apoptosis is often initiated at the mitochondria, making the mitochondria of MDR cancer cells an attractive intracellular target.

Herein, we have demonstrated via viability MTS assays and by PARP level, that encapsulating an ATM inhibitor (KU) in a previously developed TPP-functionalized drug delivery system (PTPP) is an effective means of sensitizing mammospheres to doxorubicin in a dose-dependent way (Figure 1 and Figure 4). Interestingly, we were also able to show that ATM inhibition using the PTPP–KU carrier increases DOX uptake in mammospheres (Figure 3). Drug resistance depends on mitochondria since, in general, mitochondrial function has been shown to control the susceptibility of MCF-7 and MDA-MB-231 cells to doxorubicin and paclitaxel [51]. The targeting of the mitochondrial metabolism has been shown to suppress doxorubicin resistance by controlling the drug efflux [52]. Therefore, the increased DOX uptake can tentatively be attributed to a reduced drug efflux. In line with this, it has been reported that an ATM inhibitor (AZ32) could influence multidrug resistance [53].

TPP-functionalized drug carriers are known to effectively target and be internalized by cells and organelles with large negative membrane potentials (a characteristic of rapidly proliferating malignant cells such as CSCs [33,34,35,36], as well as of mammospheres that are their close analogue). Our results are, therefore, consistent with an enhanced impact of the nanocarrier PTPP on mammosphere cells, which can be attributed to the large negative membrane potential of the mammosphere cells. Furthermore, this study demonstrates that (a) ATM has an essential role in the enhanced resistance of mammospheres to anthracycline treatment, and (b) that a mitochondriotropic nanocarrier is effective in abolishing the resistance of mammospheres to anthracycline treatment.

It has been demonstrated that ATM kinase is involved not only in DNA repair pathways, but also in the oxidative stress responses induced by several anti-cancer drugs [54], suggesting that ATM kinase plays a role in regulating mitochondrial functions. Enhanced DNA repair via efficient ATM kinase signaling is a well-known mechanism that contributes to doxorubicin resistance in cancer, but in this work, for the first time, we also described the role of mitochondria-targeting polymeric nanoparticles in sensitizing resistant cells to doxorubicin. Moreover, we hypothesize that ATM kinase activity could protect cells from DOX-induced mitochondrial damage and that directing this nanocarrier to mitochondria could reverse this function. Consequently, both increased DOX internalization and reduced DNA repair due to the inhibition of ATM kinase activity results in the observed toxicity.

This protective effect of ATM on cancer tumorsphere cells can be attributed to the induction of cell stress signaling networks through a mechanism mediated by ATM. Indeed, the nature of the main components of the mammalian stress response allows malignant tumor cells, especially upon exposure to drugs and cytotoxic conditions, to activate a phenotypic switch and pass into the tumorsphere state, whereby participating neoplastic cells take up properties of stem and progenitor cell clones, permitting at least a part of a malignant tumor to survive not only drug treatment [33], but also some “last generation” treatment schemes, including apoptosis inducers [55] and, notably, immunotherapy [56], especially therapy employing immune checkpoint inhibitors [57]. Thus, drug-induced cell stress allows malignant tumors to escape from antineoplastic treatment.

Although in normal cells doxorubicin and ATM converge on the activation of p53 [58], we here observed that, in the breast cancer cells used in this study, the impact of mitochondrial targeting, ATM inhibition, and the suppression of doxorubicin resistance were independent of the p53 status of the cells (Figure 5). This can be attributed to the multiplicity of ATM signaling networks related to the cell stress response. Even though ATM was previously shown to regulate mitophagy [14,18,33], which is a macromolecular degradation system that is involved in several major regulatory mechanisms, including the proteostatic stress response, and helps redistribute cellular materials to enable cells to adapt to adverse conditions, this is not the only role ATM plays in the cell stress response (although this may be a key role of ATM in the metabolic adaptation of mammospheres). There are other aspects of ATM’s function that also link apparently unrelated mechanisms to one another [17].

One example of this complexity can be illustrated by the activation of immune checkpoint function by the network of ATM interactions. On the one hand, it mediates the induction of transactivator nuclear factor-κB (NF-κB), leading to an increase in checkpoint inhibitor PD-L1 expression [56]. On the other hand, ATM activity is mutually dependent on PD-L1 expression, making ATM a central node between immune checkpoint function and DNA repair [59]. It must be noted that NF-κB itself can sustain breast cancer CSC, anthracycline resistance, and drug efflux, as well as the escape of cancer cells from host tissue restrictions and from several components of the immune response [60,61,62,63]. Importantly, it activates chromatin epigenetic changes that sustain these mechanisms over time, far beyond one single cell division [64]. It can therefore be expected that the inhibition of ATM impairs multiple mechanisms of survival in cancer cells, which are not limited to metabolic adaptation to cell stress, or to drug internalization, but which also include escape from the immune system and from host tissue biological surveillance.

## 5. Conclusions

In conclusion, our results have demonstrated that the encapsulation of an ATM inhibitor in a mitochondriotropic nanocarrier has the capacity to suppress doxorubicin resistance in breast cancer mammospheres independently of the cells’ p53 status. A notable effect of this combination is a substantial increase in the internalization of doxorubicin, suggesting that this strategy can potently reverse MDR in breast cancer stem cells. ATM now emerges as a pivotal regulator of breast cancer stem cell responses to stress signals, especially those signals that are induced by drugs used in antineoplastic treatment. We expect that our discovery will contribute to the development of important translational approaches to breast cancer, given the substantial therapeutic importance of breast cancer stem cells in mediating resistance to anti-cancer therapies [65]. Overall, we strongly believe that further investigation of the role of ATM kinase-dependent functions could be useful and lead to the design of new strategies for overcoming MDR in breast cancer stem cells. 

## Figures and Tables

**Figure 1 cancers-15-01474-f001:**
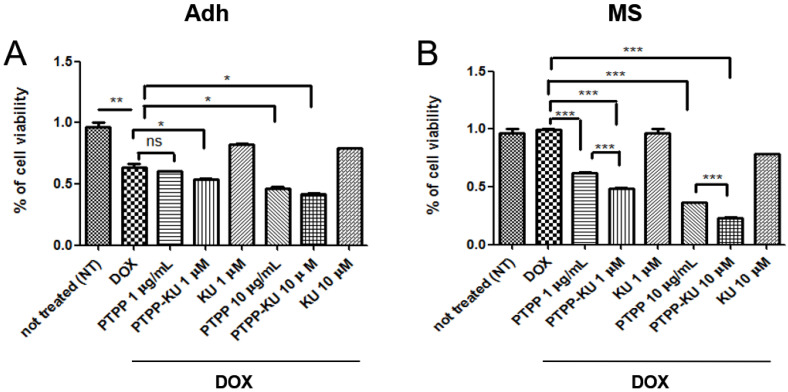
PTPP–KU sensitization of mammospheres to doxorubicin compared with non-treated (NT) cells. Cell viability of the MCF-7 cell line grown (**A**) in adherent (Adh) and (**B**) in mammosphere (MS) conditions. Cells were treated with 1 μΜ of doxorubicin (DOX) for 3 h and various doses of free KU-55933 (KU), PTPP–KU, and PTPP (administered 30 min before adding DOX), as indicated. After 24 h, cell viability was measured using an MTS assay. The results are expressed as the mean ± SD for at least three independent experiments and were analyzed using a Student’s *t*-test (* *p* < 0.05, ** *p* < 0.01, *** *p* < 0.001, ns not significant).

**Figure 2 cancers-15-01474-f002:**
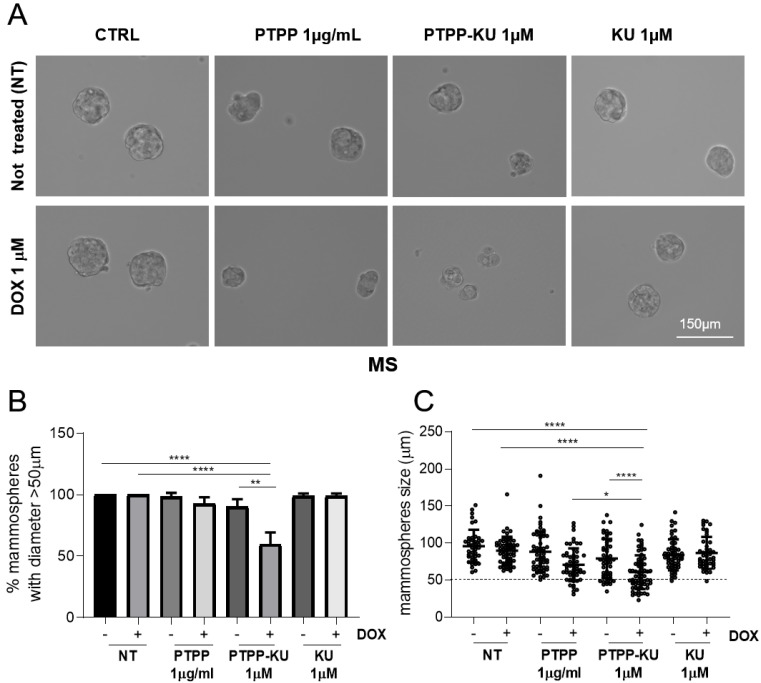
Reduction in mammosphere size caused by PTPP–KU after doxorubicin treatment compared with non-treated (NT) cells. (**A**) Optical microscopy images (20X) representative of the mammospheres’ morphology after treatment with 1 μΜ of DOX for 3 h and free KU (1 μΜ), PTPP–KU (1 μΜ), and PTPP (1 μg/mL), administered 30 min before adding DOX. Scale bar 150 µm. (**B**) Graph representing the percentage of mammospheres (diameter > 50 μm with rounded morphology) obtained from 5000 cells/mL in triplicate wells, with or without treatment. The results are expressed as the mean ± SD for at least three independent experiments and were analyzed using a Student’s *t*-test (* *p* < 0.05, ** *p* < 0.01, **** *p* < 0.0001; statistical analysis is not shown if it is not considered significant) (*n* = 4). (**C**) Graph representing the mammosphere diameters after the treatments described in (**A**).

**Figure 3 cancers-15-01474-f003:**
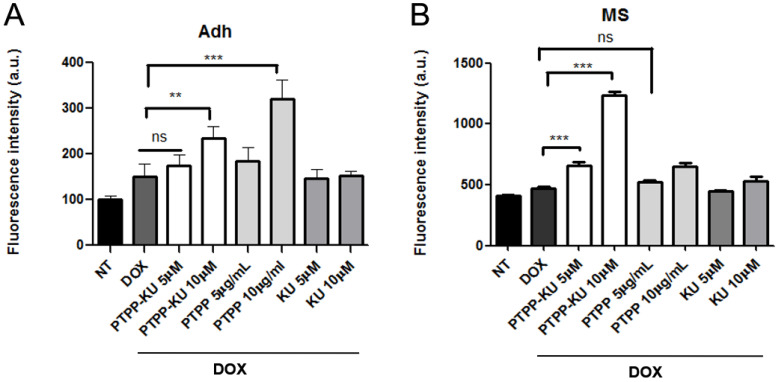
Doxorubicin internalization compared with non-treated (NT) cells. (**A**) Adherent cells (Adh) and (**B**) mammospheres (MS) in 96 well plates were treated with KU (5, 10 μM), PTPP (5, 10 μg/mL), and PTPP–KU (KU: 5, 10 μM; PTPP 5, 10 μg/mL) for 30 min before adding DOX, 1 μM. After 3 h, the wells were washed with RPMI without phenol red, and the DOX concentration was measured with an Infinite M200 plate reader (Tecan, Switzerland, λ_ex_ = 510 nm, λ_em_ = 580 nm) and expressed as fluorescence intensity in arbitrary units (a.u.). The results are expressed as the mean ± SD for at least three independent experiments and were analyzed using a Student’s *t*-test (** *p* < 0.01, *** *p* < 0.001, ns not significant). Statistical analysis is not shown if it is not considered significant (*n* = 3).

**Figure 4 cancers-15-01474-f004:**
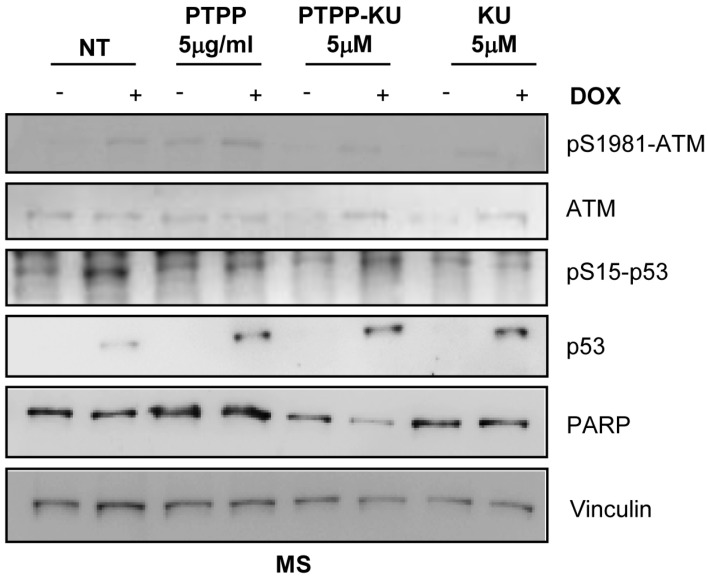
ATM activation and apoptosis induction in mammospheres co-treated with DOX and PTPP–KU. Representative Western blot of total protein extracts, performed after 24 h, from mammospheres derived from MCF-7 cells treated with DOX (1 µM) for 3 h and immunoblotted for the indicated antibodies (*n* = 3).

**Figure 5 cancers-15-01474-f005:**
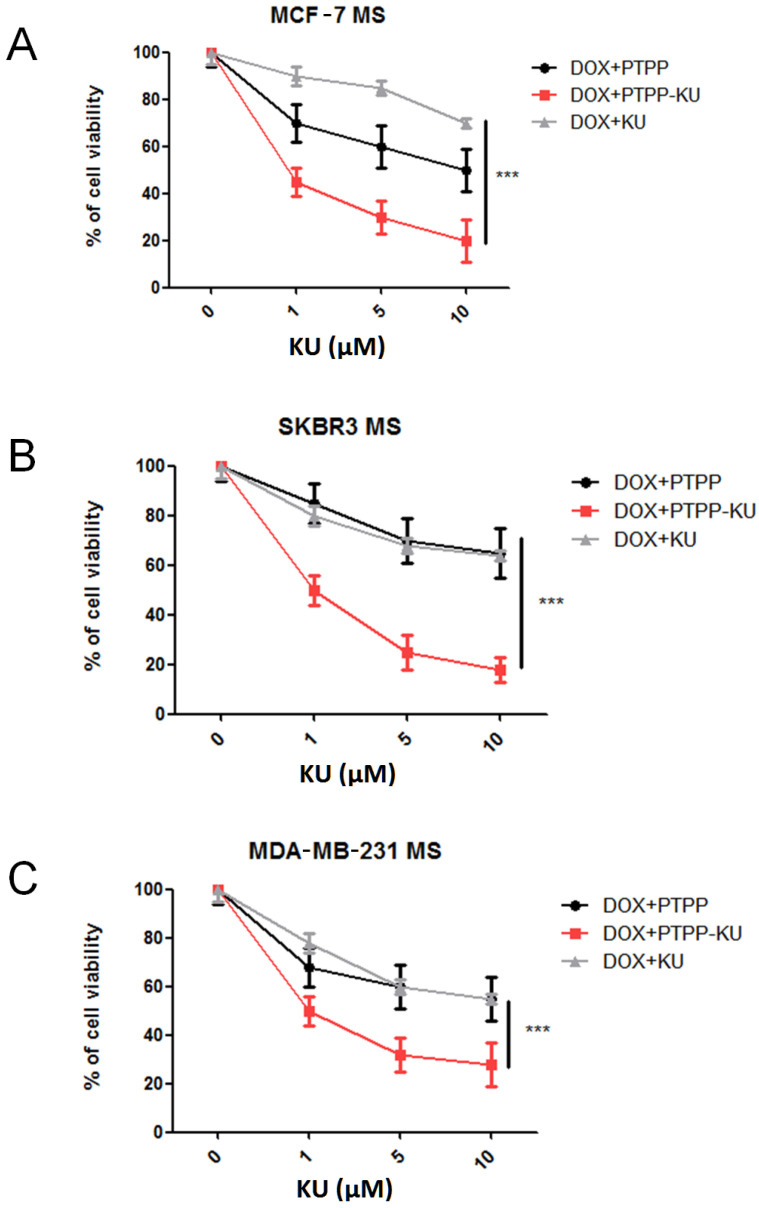
PTPP–KU sensitizes p53 WT and mutant p53 mammospheres to DOX. Cell viability of (**A**) MCF-7, (**B**) SKBR3 and (**C**) MDA-MB-231 cell lines grown in mammosphere (MS) conditions. Cells were treated with 1 μΜ of Doxorubicin (DOX) for 3 hours with increased concentrations of free KU-55933 (KU), PTPP–KU, and PTPP (administered 30 min before adding DOX), as indicated (the corresponding PTPP concentrations were 1 μg/mL, 5 μg/mL, and 10 μg/mL, respectively). After 24 h, cell viability was measured using an MTS assay. The results are expressed as the mean ± SD for at least three independent experiments and were analyzed using a Student’s *t*-test (*** *p* < 0.001) (*n* = 6).

## Data Availability

Data are contained within the article and Appendix A.

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
