# Peer review of "A Triphenylphosphonium-Functionalized Delivery System for an ATM Kinase Inhibitor That Ameliorates Doxorubicin Resistance in Breast Carcinoma Mammospheres"

_cancers, 2023, doi:10.3390/cancers15051474_

Round 1

Reviewer 1 Report

In this work, Stagni and co-workers loaded a mitochondriotropic ATM kinase inhibitor, KU-55933 in decyltriphenylphosphonium attached to polyethyleneimine (PTPP) and tested its ability to sensitize breast cancer cells and overcome the doxorubicin resistance using adherent cell culture and mammosphere models. Multiple cytotoxicity studies and a western blot study were presented.

Even though the study uses a mitochondriotropic delivery system and mitochondria-acting drugs, most of the studies presented were not mitochondria-focused. Further, there are no studies on uptake and mechanism using microscopy.

This reviewer encourages authors to conduct fluorescence microscopy-based studies to show the uptake of the PTPP-based nanoparticles in both adherent and mammosphere models and seahorse studies to check the mitochondrial effects on OXPHOS, glycolysis, etc.

Literature on mitochondria-targeted nanoparticles is not cited properly. (eg: Nano Lett. 2008, 8, 8, 2559–2563; WIREs Nanomed. Nanobiotechnol., 2015, 7, 315-329. etc)

Author Response

Reviewer 1

In this work, Stagni and co-workers loaded a mitochondriotropic ATM kinase inhibitor, KU-55933 in decyltriphenylphosphonium attached to polyethyleneimine (PTPP) and tested its ability to sensitize breast cancer cells and overcome the doxorubicin resistance using adherent cell culture and mammosphere models. Multiple cytotoxicity studies and a western blot study were presented.

Even though the study uses a mitochondriotropic delivery system and mitochondria-acting drugs, most of the studies presented were not mitochondria-focused. Further, there are no studies on uptake and mechanism using microscopy.

This reviewer encourages authors to conduct fluorescence microscopy-based studies to show the uptake of the PTPP-based nanoparticles in both adherent and mammosphere models and seahorse studies to check the mitochondrial effects on OXPHOS, glycolysis, etc.

We thank the reviewer for raising this issue, thus enabling us to depict more precisely the focus of this paper.

Our previous works verified the mitochondriotropic character of this delivery system by employing various techniques including confocal microscopy studies, mitochondrial superoxide formation, flow cytometry experiments, etc. [see References 1-3 below, corresponding to ref. 34-35 in the current manuscript]. Therefore, in this study we only focus on the investigation of the property of this system to target preferentially mammospheres of breast cancer cells. This is rationalized by the fact that this system can translocate through membranes having increased membrane potential as is the case of mammospheres that mimic cancer stem cells [see ref. 3 below corresponding to ref. 35 in the current manuscript]. It was not our intention to repeat our previous studies or to prove that KU is located in mitochondria. Accordingly, we decided to extensively modify the Title, the text in the Introduction and Discussion sections as well as the Conclusions to make our goal and findings clearer.

References

  1. Theodossiou TA, et al. Mitochondrial delivery of doxorubicin by triphenylphosphonium-functionalized hyperbranched nanocarriers results in rapid and severe cytotoxicity. Pharm Res. 2013 Nov;30(11):2832-42. doi: 10.1007/s11095-013-1111-7. Epub 2013 Aug 7. PMID: 23921486.
  2. Panagiotaki KN, et al. A Triphenylphosphonium-Functionalized Mitochondriotropic Nanocarrier for Efficient Co-Delivery of Doxorubicin and Chloroquine and Enhanced Antineoplastic Activity. Pharmaceuticals (Basel). 2017 Nov 21;10(4):91. doi: 10.3390/ph10040091. PMID: 29160846; PMCID: PMC5748647.
  3. Stagni V, et al. Targeting breast cancer stem-like cells using chloroquine encapsulated by a triphenylphosphonium-functionalized hyperbranched polymer. Int J Pharm. 2020 Jul 30;585:119465. doi: 10.1016/j.ijpharm.2020.119465. Epub 2020 Jun 1. PMID: 32497731.

Literature on mitochondria-targeted nanoparticles is not cited properly. (eg: Nano Lett. 2008, 8, 8, 2559–2563; WIREs Nanomed. Nanobiotechnol., 2015, 7, 315-329. etc)

We included the following two references according to reviewer’s suggestion.

New ref 32. Sarathi V. Boddapati, Gerard G. M. D’Souza, Suna Erdogan, Vladimir P. Torchilin, and Volkmar Weissig, Organelle-Targeted Nanocarriers: Specific Delivery of Liposomal Ceramide to Mitochondria Enhances Its Cytotoxicity in Vitro and in Vivo, Nano Lett. 2008, 8, 8, 2559–2563.

New ref 33. Rakesh K. Pathak, Nagesh Kolishetti, Shanta Dhar, Targeted nanoparticles in mitochondrial medicine, WIREs Nanomed. Nanobiotechnol., 2015, 7, 315-329.

Reviewer 2 Report

Review Report

  • A brief summary 

The present article titled “A Mitochondriotropic ATM kinase inhibitor delivery system ameliorates doxorubicin resistance in breast carcinoma mammospheres” is dealing with the overcoming breast cancer resistance to chemotherapy by using mitochondriotropic nanocarrier delivery system and ATM kinase inhibitor. The authors proved that the encapsulated inhibitor was effective against chemotherapy-resistant mammopspheres while they did not have almost any effect on adherent breast cancer cells.

  • Comments 
-The authors should explain what accounts for the toxicity of self-administration of PTPP for the both cancer cells and mammospheres (Fig. 2A and B)?

-What is the reason why PTPP-KU increased Dox sensitivity correlates with increased Dox uptake by cells?

Author Response

Reviewer 2

The present article titled “A Mitochondriotropic ATM kinase inhibitor delivery system ameliorates doxorubicin resistance in breast carcinoma mammospheres” is dealing with the overcoming breast cancer resistance to chemotherapy by using mitochondriotropic nanocarrier delivery system and ATM kinase inhibitor. The authors proved that the encapsulated inhibitor was effective against chemotherapy-resistant mammospheres, while they did not have almost any effect on adherent breast cancer cells.

 -The authors should explain what accounts for the toxicity of self-administration of PTPP for the both cancer cells and mammospheres (Fig. 2A and B)?

We thank the reviewer to give us the possibility to explain better this point. The observed toxicity of PTPP against mammospheres was extensively investigated in our previous publication [ref. 35]. Following the reviewer’s comment and in order to explain better the toxicity of PTPP we modified the relevant text (page 6, first paragraph) as follows:

Of note, we confirmed these results and we also demonstrated that PTTP alone or PTTP-KU are slightly toxic to adherent cells at all concentrations tested, while only at high concentration (10 μg/mL) show considerable toxicity (ca 70%) to MS (Supplementary Material, Figure S2). These results are consistent with previous studies designating that treatment of TPP-functionalized moieties increases cytotoxicity in cells grown in mammosphere conditions compared to cells grown in adherence [35 and a new reference 41]. The observed toxicity was attributed to the fact that PTPP is preferentially internalized in the mitochondria of mammospheres leading to mitochondrial stress increase [35].”

-What is the reason why PTPP-KU increased Dox sensitivity correlates with increased Dox uptake by cells?

We thank the reviewer to give us the possibility to clarify this point in the text.  As we discuss at the beginning of Discussion section (page 11, second and third paragraph) we have demonstrated via viability MTS assays and by detecting cleaved PARP, that encapsulating the ATM inhibitor KU to PTPP effectively sensitizes mammospheres to doxorubicin in a dose dependent way (Figures 1 and 4). In addition, we have shown that ATM inhibition using PTPP-KU nanoparticles increases DOX uptake in mammospheres (Figure 3). It is known that mitochondrial function controls the susceptibility of MCF-7 and MDA-MB-231 cells to doxorubicin [53] and that targeting of mitochondrial metabolism suppresses doxorubicin resistance by controlling drug efflux [54]. Therefore, the increased DOX uptake can tentatively be attributed to reduced drug efflux.

Moreover, as we discuss at page 11, fourth paragraph, ATM kinase activity protects cells from DOX induced mitochondrial damage and the addition of the ATM inhibitor can reduce this function. Consequently, both increased DOX internalization and reduced DNA repair due to inhibition of ATM kinase activity results in the observed toxicity.

The corresponding text at the beginning of Discussion section (page 11, second, third and fourth paragraph) has been modified accordingly.

Reviewer 3 Report

Ref: Submission ID Cancers 2080166

Comments on A Mitochondriotropic ATM kinase inhibitor delivery system 2 ameliorates doxorubicin resistance in breast carcinoma mam-3 mospheres

The study reported by Stagni et al. address an important aspect in cancer biology and is well written. However, the study is preliminary, the extensiveness of the study is below average, and the results are represented poorly with only some bar and line graphs. The study needs to be more detailed to be conclusive and there are multiple concerns in this manuscript that needs to be addressed thoroughly.

1.      Please introduce the full form of DDR in the simple summary and correct the spelling of mitochondrial in the same sentence.

2.      In line 74-76 authors mention “studies”. So please introduce more references along with reference number 19.

3.      In line 95, the term “mitochondriotropic” is introduced for the first time, please introduce, and define the term well as nothing much is mentioned about this throughout the manuscript.

4.      Clearly mention the significance of using 3 different cell lines.

5.      In Figure 1A, please provide a panel with images of cells at 20x for each mentioned time point along with the bar diagram. Also, label the distinguished features of these cells that changes with various treatments. Furthermore, mention the full form of NT in figure legends

Again, please provide magnified and clearer images for Figure1 C, clearly showing the Optical microscopy images representative of mammospheres’ morphology after treatment with DOX for 3 hours and free KU, PTPP-KU and PTPP. The present image looks blurred and unclear depicting nothing much. Also, please label the features of mammospheres.

6.      Please perform a wound healing assay on MCF-7 cell line, keeping all the treatment conditions same as Figure 1A and include as a separate figure in the manuscript. This would shed some light in the changes in the properties of cancer cells based on the different kinds of treatments.

7.      Provide the individual data points for the bar graphs shown Figure 2A and B.

8.      Also, for all the experiments undertaken, what are the sample sizes for the study and how many times were the experiments repeated? please mention this information the respective figure legends.

9.      Figures presented in the manuscript should be self-explanatory and that’s not at all the case here. In Figure 3, mention in the figure legend about the + and – groups for each type of treatment and what that signifies. Also, what is the amount of protein loaded in each well?

10.   The cleaved PARP and Ps15p53 western blots, please change the blots as they are too much postprocessed resulting in their washing out.

11.   Show images of all cell line mammospheres in 1 µM and 10µM KU samples along with line graphs for Figure 4 A,B and C.

12.   A graphical representation of the study especially showing the work flow and important findings the authors have highlighted in their study should be included for better understanding of the study.

Author Response

Reviewer 3

The study reported by Stagni et al. address an important aspect in cancer biology and is well written. However, the study is preliminary, the extensiveness of the study is below average, and the results are represented poorly with only some bar and line graphs. The study needs to be more detailed to be conclusive and there are multiple concerns in this manuscript that needs to be addressed thoroughly.

We thank the reviewer for his/hers comments that help us improve the quality of the manuscript.

  1. Please introduce the full form of DDR in the simple summary and correct the spelling of mitochondrial in the same sentence.

We thank the reviewer for highlighting to us that the construction of the Simple Summary was not clear. In the revised version the Simple Summary was extensively modified and the spelling errors were corrected.

  1. In line 74-76 authors mention “studies”. So please introduce more references along with reference number 19.

According to the reviewer’s suggestion we added the following references:

New ref 20. Phenformin and ataxia-telangiectasia mutated inhibitors synergistically co-suppress liver cancer cell growth by damaging mitochondria FEBS Open Bio. 2021 May;11(5):1440-1451. doi: 10.1002/2211-5463.13152

New ref 21. Sullivan KD, Palaniappan VV, Espinosa JM. ATM regulates cell fate choice upon p53 activation by modulating mitochondrial turnover and ROS levels. Cell Cycle. 2015;14(1):56-63. doi: 10.4161/15384101.2014.973330. 

New ref 22. Ataxia-telangiectasia mutated kinase regulates ribonucleotide reductase and mitochondrial homeostasis. J Clin Invest. 2007 Sep;117(9):2723-34. doi: 10.1172/JCI31604.

  1. In line 95, the term “mitochondriotropic” is introduced for the first time, please introduce, and define the term well as nothing much is mentioned about this throughout the manuscript.

We modified the text (page 2, last paragraph) as follows:

“A number of mitochondriotropic moieties, i.e. moieties that can target mitochondria, have already been identified....”

  1. Clearly mention the significance of using 3 different cell lines.

We thank the reviewer for raising this issue. We used three different cell lines to examine whether breast cancer cell lines expressing mutant p53 give a different response compared to the wild-type p53 MCF-7 cells. To this aim, we also studied MDA-MB-231 cells that express the tumor suppressor gene p53 mutant R280Kand also SK-BR-3 cells, which express the mutant form of tumor suppressor p53, R175H. The manuscript was modified accordingly as shown below (page 10, first paragraph):

“To evaluate the necessity for intact p53 function, we also utilized cell lines with mutant p53. For this aim, we performed viability experiments in mammospheres derived from p53 WT cells (MCF-7) or mutant p53 cells (SKBR3 and MDA-MB-231 cells).”

  1. In Figure 1A, please provide a panel with images of cells at 20x for each mentioned time point along with the bar diagram. Also, label the distinguished features of these cells that changes with various treatments. Furthermore, mention the full form of NT in figure legends. Again, please provide magnified and clearer images for Figure1 C, clearly showing the Optical microscopy images representative of mammospheres’ morphology after treatment with DOX for 3 hours and free KU, PTPP-KU and PTPP. The present image looks blurred and unclear depicting nothing much. Also, please label the features of mammospheres.

We thank the reviewer for raising this issue, thus enabling us to add an important piece of information to our manuscript. We addressed it by performing again the experiments in triplicate and by taking 20X zoom photographs at different time points. We labelled mammospheres counting only the mammospheres with spheroid morphology over 50 μm size after 24 h, i.e. at the same time point in which we measure cell viability, and after 3h . We added all relevant information in the caption of the new Figure 2 and in the new Figure S3.

  1. Please perform a wound healing assay on MCF-7 cell line, keeping all the treatment conditions same as Figure 1A and include as a separate figure in the manuscript. This would shed some light in the changes in the properties of cancer cells based on the different kinds of treatments.

We thank the reviewer for this suggestion, but this experiment can only be performed in adherent cells, not in mammospheres. Therefore, we would not be able to have a direct comparison between adherent cells vs. mammospheres with this experiment; thus, this experiment will not provide any further proof on whether we can preferentially target mammospheres.

  1. Provide the individual data points for the bar graphs shown Figure 2A and B.

We added the raw data of this experiment (Fig. 2 is now Fig. 3) in the Supplementary Material file (Figure S4).

  1. Also, for all the experiments undertaken, what are the sample sizes for the study and how many times were the experiments repeated? please mention this information the respective figure legends.

We thank the reviewer for the suggestion. We add the requested information in the respective Figure captions.

  1. Figures presented in the manuscript should be self-explanatory and that’s not at all the case here. In Figure 3, mention in the figure legend about the + and – groups for each type of treatment and what that signifies. Also, what is the amount of protein loaded in each well?

We thank the reviewer for this suggestion. We decided to perform a new Western blot in triplicate (Figure 4). The original blots can be found in Supplementary Material (Figure S5).

  1. The cleaved PARP and Ps15p53 Western blots, please change the blots as they are too much postprocessed resulting in their washing out.

As mentioned above we performed again these experiments in triplicate (n=3) and we added a new Figure 4 with all relevant experimental information.

  1. Show images of all cell line mammospheres in 1 µM and 10 µM KU samples along with line graphs for Figure 4 A,B and C.

As shown in Figure 5 (which is the old Figure 4), all three cell lines are responding almost identically to PTPP-KU+DOX and therefore it is not expected that the images of SKBR3 or MDA-MD-231 would be substantially different that those of MCF-7.

  1. A graphical representation of the study especially showing the work flow and important findings the authors have highlighted in their study should be included for better understanding of the study.

We added a graphical abstract as requested.

Reviewer 4 Report

This study by Stagni et al utilizes a nanoparticle (PTPP) delivery system to combintorially treat breast cancer spheres with DOX and the ATM inhibitor, KU. The authors assert that this delivery system sensitizes cells to DOX treatment. 

Overall, while I believe the authors need to solidify the way they present the study conceptualization and data interpretation, I do believe that the data are worth publication. In order for this study to be acceptable for publication in this journal, the authors need to address the following items:

1) While statistical significance is achieved in reducing cell viability in Fig 2B with when KU is introduced with PTPP, the actual decrease in viability of DOX+PTPP+KU compared to DOX+PTPP is only a few percent. With the high DOX concentration used in all of these experiments (1 micromolar), it is difficult to assess whether KU (with PTPP) really is sensitizing cell viability. 

2) Figure 2B demonstrates that PTPP does not enhance DOX uptake in MS, yet Figure 1B demonstrates that PTPP induces a 40% decrease in viability in MS compared to DOX alone. Together, these data may indicate that PTPP itself. The experiments in Figures 1 and 2 do not show PTPP alone, and therefore are missing an important control. 

3) The experiments above also indicate that (as the authors suggest in the results text) that PTPP+KU increase DOX uptake. They should provide some assessment as to a possible mechanism for this. 

4) Related to the comment above, the authors need to justify their contention that PTPP is actually targeting DOX and KU to mitochondria. It is very possible that PTPP + KU + DOX is simply getting more drugs into the cytosol or other compartments. There is no evidence in this study that mitochondria actually have anything to do with the data or conclusions presented here. 

5) Also related to the above comments, the authors need to provide a rational explanation for targeting KU to mitochondria. Many signaling proteins impact mitochondrial function and mitophagy in a similar manner to ATM without ever actually localizing to mitochondria. Therefore, the apparent hypothesis of the authors (ATM impacts mitochondrial function and therefore we should target ATP inhibitors to mitochondria) is completely without basis. The authors should either cite studies that have clearly demonstrated ATM localization to mitochondria or demonstrate this themselves. Without this evidence, it is difficult to take this study seriously as having anything to do with mitochondria. Alternatively, as the authors indicated, KU inhibits other enzymes and therefore may be inhibiting non-ATM mitochondrially-localized enzymes. 

Author Response

Reviewer 4

This study by Stagni et al utilizes a nanoparticle (PTPP) delivery system to combintorially treat breast cancer spheres with DOX and the ATM inhibitor, KU. The authors assert that this delivery system sensitizes cells to DOX treatment. 

Overall, while I believe the authors need to solidify the way they present the study conceptualization and data interpretation, I do believe that the data are worth publication. In order for this study to be acceptable for publication in this journal, the authors need to address the following items:

1) While statistical significance is achieved in reducing cell viability in Fig 2B with when KU is introduced with PTPP, the actual decrease in viability of DOX+PTPP+KU compared to DOX+PTPP is only a few percent. With the high DOX concentration used in all of these experiments (1 micromolar), it is difficult to assess whether KU (with PTPP) really is sensitizing cell viability. 

We thank the reviewer for this comment that gave us the opportunity to clarify this point. The employed DOX concentration is higher that normally used in other studies because the cell lines used in this study are DOX resistant.  As shown in Fig. 1B, 1 μM of DOX is nontoxic (100% viability) for mammospheres and this is the reason why we did not employ lower DOX concentrations. The effect of DOX+PTPP or DOX+PTPP+KU on MS, even when KU=1 μM, is clearly shown in this figure (p=***) (the actual values are ~60% and ~45% viability, respectively).

2) Figure 2B demonstrates that PTPP does not enhance DOX uptake in MS, yet Figure 1B demonstrates that PTPP induces a 40% decrease in viability in MS compared to DOX alone. Together, these data may indicate that PTPP itself. The experiments in Figures 1 and 2 do not show PTPP alone, and therefore are missing an important control. 

The toxicity of PTPP against mammospheres was extensively investigated in our previous publication (ref. 35) and for this reason the effect of PTPP alone in this manuscript was only presented in the Supporting Information (Fig. S2), where it is clear that PTPP alone confers toxicity only at high concentrations (10 μg/mL) to mammospheres. According to the reviewer’s suggestion we modified this part of the manuscript (page 6, first paragraph) to make clear the effect of PTPP alone (without DOX) as follows:

 “Of note, we confirmed these results and we also demonstrated that PTTP alone or PTTP-KU are slightly toxic to adherent cells at all concentrations tested, and only at high concentration (10 μg/mL) show considerable toxicity (ca 70%) to MS (Supplementary Material, Figure S2). These results are consistent with previous studies designating that treatment of TPP-functionalized moieties increases cytotoxicity in cells grown in mammosphere conditions compared to cells grown in adherence [35 and a new ref]. The observed toxicity was attributed to the fact that PTPP is preferentially internalized in the mitochondria of mammospheres leading to mitochondrial stress increase [35].”

3) The experiments above also indicate that (as the authors suggest in the results text) that PTPP+KU increase DOX uptake. They should provide some assessment as to a possible mechanism for this.

We thank the reviewer to give us the possibility to clarify this point in the text.  As we discuss at the beginning of Discussion section (page 11, second and third paragraph) we have demonstrated via viability MTS assays and by detecting cleaved PARP, that encapsulating the ATM inhibitor KU to PTPP effectively sensitizes mammospheres to doxorubicin in a dose dependent way (Figures 1 and 4). In addition, we have shown that ATM inhibition using PTPP-KU nanoparticles increases DOX uptake in mammospheres (Figure 3). It is known that mitochondrial function controls the susceptibility of MCF-7 and MDA-MB-231 cells to doxorubicin [53] and that targeting of mitochondrial metabolism suppresses doxorubicin resistance by controlling drug efflux [54]. Therefore, the increased DOX uptake can tentatively be attributed to reduced drug efflux.

The corresponding text at the beginning of Discussion section (page 11, second and third paragraph) has been modified accordingly.

4) Related to the comment above, the authors need to justify their contention that PTPP is actually targeting DOX and KU to mitochondria. It is very possible that PTPP + KU + DOX is simply getting more drugs into the cytosol or other compartments. There is no evidence in this study that mitochondria actually have anything to do with the data or conclusions presented here. 

Our intention was to provide evidence that this system is effective against mammospheres, which are considered analogues of cancer stem cells. We do not claim that KU or DOX are targeting mitochondria. In fact, specifically for DOX, which is not encapsulated in PTPP, there is no reason to be localized in mitochondria. We agree with the reviewer that PTPP + KU is simply getting more DOX into cells and we deleted any text about targeting drugs (DOX or KU) to mitochondria.

5) Also related to the above comments, the authors need to provide a rational explanation for targeting KU to mitochondria. Many signaling proteins impact mitochondrial function and mitophagy in a similar manner to ATM without ever actually localizing to mitochondria. Therefore, the apparent hypothesis of the authors (ATM impacts mitochondrial function and therefore we should target ATP inhibitors to mitochondria) is completely without basis. The authors should either cite studies that have clearly demonstrated ATM localization to mitochondria or demonstrate this themselves. Without this evidence, it is difficult to take this study seriously as having anything to do with mitochondria. Alternatively, as the authors indicated, KU inhibits other enzymes and therefore may be inhibiting non-ATM mitochondrially-localized enzymes. 

In the revised version we modified the Title, Abstract, Introduction and Conclusions sections to make it clear that we do not claim that KU localized mitochondria, but that when KU is encapsulated in PTPP becomes more potent towards mammospheres and increases DOX uptake and toxicity.

Round 2

Reviewer 1 Report

In light of the change in title and the explanation provided by the authors, the current version is suitable for publication after minor revisions.

1) the old western blot data from the original submission is very different from the new one. Authors should provide both the data in the revised manuscript and explain why the differences.

2) Authors did not justify why suddenly multiple additional authors were added to the manuscript, just before its acceptance.

Author Response

In light of the change in title and the explanation provided by the authors, the current version is suitable for publication after minor revisions.

  • the old western blot data from the original submission is very different from the new one. Authors should provide both the data in the revised manuscript and explain why the differences.

Thank you for the suggestion. We show below, for the reviewer, the quantification of the originally submitted multiple Western blot (n=4). Overall, the data are not so different from the Western blot presented in Figure 4: there is a downregulation of ATM phosphorylation in all experiments especially for mammospheres treated with KU and PTPP-KU. The downregulation of total PARP is the same in all Western blots, as well as the downregulation of phospho-S15 p53. Therefore, the conclusions are the same, reappearing in the blot several times.

  • Authors did not justify why suddenly multiple additional authors were added to the manuscript, just before its acceptance

To satisfy reviewer comments we have to perform new mammosphere experiments. For this reason, we have to ask the help of Prof. Barilà’s laboratory with the help of her post-doc Dr Claudia Contadini, and share with us expertise and reagents, because this type of experiments is very expensive and time consuming. This is due to the fact that Dr Rosario Luigi Sessa is no longer in the laboratory and he could not perform new experiments with mammospheres. We explain this to the editor in the cover letter.

Reviewer 3 Report

The authors have addressed most if the suggestions and comments satisfactorily. So, it can be accepted for publication. However, the authors said in response to one of my comments that "We added a graphical abstract as requested." But I couldn’t find the figure in the revised main file or the supplementary one. Can you please provide that in the final paper? Thanks

Author Response

The authors have addressed most if the suggestions and comments satisfactorily. So, it can be accepted for publication. However, the authors said in response to one of my comments that "We added a graphical abstract as requested." But I couldn’t find the figure in the revised main file or the supplementary one. Can you please provide that in the final paper? Thanks.

We uploaded the graphical abstract as a separate file. We attach it also here
